# Potential Source and Transmission Pathway of Gut Bacteria in the Diamondback Moth, *Plutella xylostella*

**DOI:** 10.3390/insects14060504

**Published:** 2023-05-31

**Authors:** Shuncai Han, Qianqian Ai, Xiaofeng Xia

**Affiliations:** 1State Key Laboratory of Ecological Pest Control for Fujian and Taiwan Crops, Fujian Agriculture and Forestry University, Fuzhou 350002, China; 2Institute of Applied Ecology, Fujian Agriculture and Forestry University, Fuzhou 350002, China; 3Key Laboratory of Integrated Pest Management for Fujian-Taiwan Crops, Ministry of Agriculture and Rural Affairs, Fuzhou 350002, China

**Keywords:** *Plutella xylostella*, gut bacteria, source, transmission mode

## Abstract

**Simple Summary:**

*Plutella xylostella* is a major pest of Cruciferae vegetables all over the world. Gut bacteria play an important role in the life activities of *P. xylostella*, but so far, little is known about the source and transmission of gut bacteria of *P. xylostella*. Therefore, we used the traditional microbial culture method to show that there is a potential correlation between the gut bacteria of *P. xylostella* and food bacteria, and *P. xylostella* gut bacteria exhibit vertical and horizontal transmission through eggs. Our research results will contribute to biological pest control based on gut bacteria.

**Abstract:**

*Plutella xylostella* (L.), commonly known as the diamondback moth, is currently a major worldwide pest. Gut bacteria play an important role in the physiology and insecticide resistance of *P. xylostella*, but little is known about the sources and transmission routes of its gut bacteria. In this study, traditional microbial culture methods were used to analyze the sources and transmission modes of gut bacteria in *P. xylostella*, which could help develop pest control strategies based on gut bacteria. The main findings are as follows: gut bacterial diversity was significantly higher in *P. xylostella*-fed radish sprouts than those fed an artificial diet, indicating a potential association between gut bacteria and food bacteria. In addition, sequence analysis revealed the isolation of *Enterobacter* sp., *Pantoea* sp., *Cedecea* sp., and *Pseudomonas* sp. from both radish sprouts and *P. xylostella*. Importantly, *Enterobacter* sp. was found in all tested samples (radish sprouts, gut of *P. xylostella*, ovaries, and eggs), suggesting that bacteria acquired from food could be transferred from the gut to the ovaries and eggs. This was confirmed through experiments, which also showed that eggs could carry bacteria and transmit them to the gut, indicating vertical transmission of gut bacteria via eggs. Furthermore, the 3rd instar larvae of *P. xylostella* with and without gut bacteria were mixed and raised until the 4th instar. Then, we found that all the gut of the 4th instar larvae carried the same bacteria, indicating that the gut bacteria of *P. xylostella* can be horizontally transmitted through social behavior. This study lays a foundation for further exploration of the sources, transmission, and coevolution of the host of gut bacteria in *P. xylostella*, and provides new insights into pest control strategies based on the source and transmission of gut bacteria.

## 1. Introduction

Insects have a complex symbiotic relationship with gut bacteria, which help the hosts digest food [1], resist parasites and pathogens [2,3], facilitate inter-species communication [4], and regulate mating and reproductive systems [5,6]. Therefore, research on insect gut bacteria is particularly important in the field of plant protection. The structure of insect gut bacteria is influenced by the type of food they consume. For example, *Diaphorina citri* has significant differences in gut bacteria among different hosts, with the highest gut bacteria diversity found in insects that feed on *Citrus poonensis* cv. Ponkan and the lowest diversity in those that feed on *Citrus reticulata* cv. Shatangju [7]. Host plants have a significant impact on the structure and composition of gut bacteria in *Spodoptera frugiperda* [8].

It is worth noting that insects have evolved various mechanisms to vertically transmit beneficial bacteria to their offspring or horizontally spread them within and between populations [9]. Studies have shown that gut bacteria *Snodgrassella alvi* and *Gilliamella apicola* in field *Bombus terrestris* populations can be vertically transmitted from mothers to offspring [10]. *Serratia symbiotica*, a gut bacterium in *Aphidoidea*, can also be transmitted vertically from mothers to offspring [11]. Social insects, such as *Cryptocercus* sp., *Reticulitermes speratus,* and *Apis mellifera*, which engage in trophallaxis or coprophagy, can directly or indirectly facilitate the horizontal transmission of gut bacteria, promoting the coevolution of host insects with their gut bacteria [12,13,14]. In an experiment where 20 newly emerged bees and 20 older worker bees from the same hive but marked with different colored paint were mixed and fed with bee bread in a cage, characteristic bacteria were detected in the gut of the newly emerged bees [14], indicating that the gut bacteria of bees can be horizontally transmitted through social activity within the population. *Wolbachia* of *Homalotylus* is also capable of horizontal transmission between populations [15].

The diamondback moth *Plutella xylostella* (L.) (Lepidoptera: Plutellidae), is a major pest of cruciferous vegetables distributed worldwide [16,17]. The life cycle of *P. xylostella* includes egg, larva, pupa, and adult, with the larvae consisting of four instars. Early studies investigated the abundance and diversity of gut bacteria in *P. xylostella* at different developmental stages [18]. Subsequently, detailed studies were conducted on gut bacteria in *P. xylostella* populations collected from different geographic regions in India, revealing that gut bacteria of *P. xylostella* are influenced by different geographic regions, which may be due to changes in latitude, environmental factors, and the insect’s adaption to its local climate [19]. However, research has shown that both environmental factors and food sources have an impact on the diversity of insect gut bacteria [20]. Our previous research studied the composition of gut bacteria in *P. xylostella* [21], its functional relationship with host feeding [22], and its relationship with insecticide resistance [23]. However, little is known about the source and transmission mode of gut bacteria in *P. xylostella*. In this study, we aim to analyze the potential correlation between gut bacteria in *P. xylostella* and food bacteria, study its vertical and horizontal transmission, lay a foundation for further research on gut bacteria in *P. xylostella*, and provide ideas for controlling *P. xylostella* based on the source and transmission of gut bacteria. 

## 2. Materials and Methods 

### 2.1. Feeding P. xylostella

*P. xylostella* used in this study was from the Institute of Zoology, the Chinese Academy of Sciences, and was domesticated by feeding on an artificial diet. The artificial diet consisted of 6 g of agar mixed with 250 mL of ddH_2_O, heated in a microwave until fully dissolved, and cooled to about 70 °C. Then, 37.5 g of wheat bran, 20 g of yeast powder, 10 g of sucrose, 3 g of radish seeds, 0.8 g of compound vitamins, 1 g of citric acid, 1 g of nipagin, and 1 g of vitamin C were added, followed by 1 mL of rapeseed oil and 50 μL of linoleic acid. The mixture was stirred well before use. Feeding *P. xylostella* separately with an artificial diet and radish sprouts resulted in two strains: the artificial diet strain and the radish sprout strain. The larvae possess four instars in both diets. The artificial climate room for rearing larvae was maintained at a temperature of 25 ± 1 °C, a relative humidity (RH) of 40–70%, and a light/dark photoperiod (L:D) of 16:8 h. Radish sprouts, fed upon by the larvae, were grown in an artificial climate room with a temperature of 23 ± 1 °C and an RH of approximately 75%. The variety of radish seeds used was “Spring full ground Nanpanzhou Daiko” (Fuzhou Yongrong Seed Co., Ltd., Fuzhou, China). The artificial climate room for rearing the larvae on radish sprouts was maintained at a temperature of 25 ± 1 °C, an RH of 70–80%, and an L:D photoperiod of 16:8 h. Adult moths were provided with 10% honey water as food. 

### 2.2. Isolation, Culture, and Identification of Symbiotic Bacteria 

#### 2.2.1. Isolation and Culture of Bacteria from Radish Sprouts 

Radish sprouts were planted using horticultural universal cultivation soil (Rongfeng Horticulture Company, Guangzhou, China). After 3 days, the seeds germinated, and after 7 days, ten radish leaves were randomly selected (each leaf was approximately 87 mm^2^ in size). The leaves were placed in 2 mL centrifuge tubes with 200 μL of sterile water, and crushed using a pipette tip, two sterilized steel beads were added to each tube, then the tubes were shaken until the leaves were completely dissolved. The culture media include Luria-Bertani medium (LB) [24], nutrient agar (NA) [24], and anaerobic agar (mixing 20.0 g of pancreatic digest of casein, 5.0 g of sodium chloride, 10.0 g of dextrose, 1.0 g of sodium formaldehyde sulfoxylate, 2.0 g of sodium thioglycolate, 0.002 g of aniline blue water soluble, 20 g of agar with 1 L of distilled water, 7.2 of potential hydrogen (pH), then sterilized by high-pressure steam at 121 °C for 30 min after being packaged and sealed). Ten-fold serial (10^−1^, 10^−2^, and 10^−3^) dilutions of dissolved solution of radish leaves were plated on LB and NA mediums. The stock solution and ten-fold serial (10^−1^ and 10^−2^) dilutions of dissolved solution of radish leaves were plated on anaerobic agar medium. An amount of 20 μL of each dilution mentioned above was spread onto the culture media, repeated 3 times, and incubated in a 37 °C incubator for 96 h. Isolation and cultivation of bacteria: Individual bacterial colonies with different sizes, colors, and morphologies were isolated and purified five times on LB mediums using an inoculation loop to obtain single clones (Appendix A). After purification, the bacterial strains were cultured in liquid LB mediums and preserved with 25% glycerol at −80 °C. 

#### 2.2.2. Isolation and Culture of Gut Bacteria from *P. xylostella*

Dissection: 30 healthy 4th instar larvae (the *P. xylostella* start feeding heavily from the beginning of the 4th instar), 30 pupae, and 30 adults of *P. xylostella* were randomly selected for dissection. The 4th instar larva is a representative larval stage of *P. xylostella*. Due to a large amount of feeding, there are abundant microorganisms in the gut, and this stage is at the end of the larval stage, making it easier to compare its association with the gut microbiota of pupae and adults. Meanwhile, the insect body is large and easy to dissect; thus, the 4th instar larvae were chosen for study. Before dissection, the adults were frozen at −20 °C for 5 min to immobilize them. The selected insects (the 4th instar larvae, pupae. and adults) were dissected on a UV-sterilized and ultra-clean workbench. Their bodies were washed with sterile water, surface-sterilized with 75% ethanol for 1 min, and then washed again with sterile water. The isolated gut tissues were put into 2 mL centrifuge tubes containing 200 μL sterile water, crushed with a pipette tip, and then two sterilized steel beads were added and shaken until the gut tissues were completely dissolved. The LB, NA, and anaerobic agar mediums were used for bacterial culture. Both ten-fold serial (10^−3^, 10^−4^, and 10^−5^) dilutions of the dissolved solution of larval guts and ten-fold serial (10^−2^, 10^−3^, 10^−4^, and 10^−5^) dilutions of the dissolved solution of pupal guts were plated on LB, NA, and anaerobic agar mediums. Ten-fold serial (10^−2^, 10^−3^, and 10^−4^) dilutions of the dissolved solution of adult guts were plated on LB and NA mediums. Ten-fold serial (10^−1^, 10^−2^, and 10^−3^) dilutions of the dissolved solution of adult guts were plated on anaerobic agar medium. An amount of 10 μL of each dilution mentioned above was spread onto the culture medium, repeated 3 times, and incubated in a 37 °C incubator for 96 h (Appendix A). Bacterial isolation and cultivation conditions were the same as in Section 2.2.1 (Appendix A).

#### 2.2.3. Isolation and Culture of Bacteria from the Ovary of *P. xylostella*

Ten healthy female *P. xylostella* were randomly selected, and the dissection was the same as in Section 2.2.2. The stock solution and ten-fold serial (10^−1^ and 10^−2^) dilutions of the dissolved solution of ovaries were plated on LB, NA, and anaerobic agar mediums. An amount of 10 μL of each dilution mentioned above was spread onto the culture medium, repeated 3 times, and incubated in a 37 °C incubator for 96 h (Appendix A). The isolation and culture conditions of bacteria were the same as in Section 2.2.1 (Appendix A), and the experiment was repeated three times.

#### 2.2.4. Isolation and Culture of Bacteria from Eggs of *P. xylostella*

A new oviposition card was placed in the adult rearing cage. After 30 min, 200 eggs of *P. xylostella* were collected and placed in a 2 mL centrifuge tube with 200 μL sterile water. Two sterilized steel beads were added to the tube, which was then shaken until the eggs were completely dissolved. Ten-fold serial (10^−2^, 10^−3^, 10^−4^, and 10^−5^) dilutions of the dissolved solution of eggs were plated on LB, NA, and anaerobic agar mediums. An amount of 20 μL of each dilution mentioned above was spread onto the culture medium, repeated 3 times, and incubated in a 37 °C incubator for 96 h (Appendix A). The isolation and culture conditions of bacteria are the same as in Section 2.2.1 (Appendix A), and the experiment was repeated three times.

#### 2.2.5. Identification of Bacteria 

DNA was extracted from isolated and purified bacteria. DNA Extraction Kit: TaKaRa MiniBEST Bacterial Genomic DNA Extraction Kit Ver.3.0 (TaKaRa Biomedical Technology (Beijing) Co., Ltd., Beijing, China). Amplification was performed as previously described [24]. DNA was amplified using universal primers (27 F, 1492 R). DNA polymerase: Phanta Max Super-fidelity DNA Polymerase (Nanjing Vazyme Biotech Co., Ltd, Nanjing, China). Then, amplified DNA was sent to Boshang Biological Corporation (Shanghai, China) for sequencing, the results were compared by blast, and phylogeny was compared.

### 2.3. Vertical Transmission of Gut Bacteria of P. xylostella 

#### 2.3.1. Tracing Gut Bacterial Transmission by Resistant Bacteria

1. Preparation of anti-kanamycin *Enterobacter* sp. RE1-kN: (1) The competent cells of *Enterobacter* sp. RE1 (GenBank Access Number: MH141495) were prepared, and the anti-kanamycin GFP plasmid (PET28a-EGFP plasmid (Miaoling Biotechnology Corporation, Wuhan, China)) was introduced into them. (2) Some single colonies of anti-kanamycin *Enterobacter* sp. RE1-KN were selected, and they were shaken at 37 ℃ and 200 rpm overnight, then the shaken bacterial suspension was poured into a sterile 50 mL centrifuge tube in an ultra-clean workbench sterilized by UV for 30 min, and centrifuged for 10 min at 5000 rpm in a high-speed refrigerated centrifuge. (3) The above-mentioned centrifuge tube was shaken on a vortex mixer until the precipitate was dispersed, and then sterile water was added. After shaking evenly, the mixture was centrifuged at 5000 rpm for 10 min, and the supernatant was discarded. This washing step was repeated three times. (4) Then, the above-mentioned centrifuge tube was filled with sterile water, mixed evenly, and diluted the original solution 7 times, then the OD600 value was measured using a UV–Visible spectrophotometer, and the OD600 value of the original solution was calculated.

2. Rearing *P. xylostella*: (1) 40 mL sterile water and 40 μL of 50 mg/mL kanamycin were added to a sterile glass bottle. (2) An amount of 100 mL artificial diet was poured into a disposable culture dish and cut into 2 × 2 cm square pieces with a blade, put into kanamycin solution, soaked for 30 min, and then they were dried. (3) The 3rd instar larvae were fed with a diet soaked in kanamycin solution for 24 h, and then they were fed with a diet soaked in *Enterobacter* sp. RE1-KN solution (OD600 = 2.0) for 30 min. 

3. Detection: (1) *P. xylostella* fed on a diet containing *Enterobacter* sp. RE1-KN, then LB solid mediums containing kanamycin were coated with the gut solution of the *P. xylostella* (4th instar larvae, pupae, and adults), adult ovaries, and the sterile water, which soaked eggs (The eggs of *P. xylostella* feeding on the diet containing *Enterobacter* sp. RE1-KN were brushed onto a sterile weighing paper with a sterile bristle brush. Then, the eggs were placed in a centrifuge tube, and sterile water was added). The plates were sealed and incubated upside down at 37 °C for 12 h. (2) PCR detection: Some single colonies were selected and put into a 1.5 mL centrifuge tube containing 20 μL Elution Buffer (Nanjing Vazyme Biotech Co., Ltd., Nanjing, China) that can elute PET28a-EGFP plasmid from *Enterobacter* sp. RE1-KN, then it was heated in a water bath at 95 °C for 10 min. After centrifugation, 1 μL of supernatant was obtained, and PCR amplification was performed using T7 primer selection system (Appendix A) and procedure (Appendix A).

#### 2.3.2. Detection of Egg Surface Bacteria Entering the Gut

1. (1) The sterile artificial diet was sub-packed into a conical flask and dried on an ultra-clean workbench. Sterile artificial diet: 15 g wheat germ powder, 8 g yeast powder, 4 g sucrose, 2.4 g agar, 1.2 g radish seeds, and 100 mL pure water was added into a 250-mL conical flask, then 400 μL rapeseed oil and 25 μL linoleic acid was added. After mixing well, the mouth of the conical flask was wrapped with 8 layers of medical degreased gauze and sealed with sealing film, then the conical flask was sterilized at 115 °C under high-pressure steam for 30 min. Afterward, the mixture (0.032 g multivitamin, 0.04 g sorbic acid, 0.04 g nipagin, 0.04 g Vitamin C, and 5 mL pure water) was filtered and sterilized through a microporous filter film with a pore size of 0.23 μm before being added. (2) The sealing film containing eggs of *P. xylostella* was washed once with sterile water, sterilized with 1.5% sodium hypochlorite for 15 s, then washed twice with sterile water and dried (Sterile water used to clean the sealing film containing eggs of *P. xylostella* for the last time was coated on LB solid mediums to test whether the sterilization was complete.), as the control group. The treatment group was left to soak the *Enterobacter* sp. RE1-KN solution (OD600 = 2.0) for 30 min and dry. (3) The sealing films containing eggs of the control group and the treatment group were put into the glass culture bottles containing sterile artificial diet, each bottle mouth was wrapped with sterile 8-layer medical absorbent gauze, sealed with a rubber band, and then the culture bottles were tilted for cultivation.

2. Detection (1) LB solid mediums containing kanamycin were coated with the gut solution of 4th instar larvae in the treatment group and control group, which were sealed and put into an incubator at 37 °C. After 12 h, the LB solid mediums were observed for colony growth. (2) PCR detection was the same as in Section 2.3.1.

### 2.4. Horizontal Transmission of Gut Bacteria of P. xylostella 

(1) The initial 3rd instar larvae were selected and starved for 12 h. (2) The *P. xylostella* in the control group were fed with a normal diet after starvation, and the *P. xylostella* in the treatment group were fed with a diet soaked in *Enterobacter* sp. RE1-KN solution (OD600 = 2.0) for 30 min. The *P. xylostella* in both the control and treatment groups was raised for 24 h. (3) 5 larvae in the treatment group and 5 larvae in the control group were placed in the same new insect-rearing box and fed with a normal artificial diet (4 repetitions). Fresh diet was changed once a day and the gut of 4th instar larvae were dissected. (4) PCR detection of gut bacteria was the same as in Section 2.3.1.

## 3. Results 

### 3.1. Isolation and Identification of Bacteria from Radish Sprouts

In this study, 24 strains of different bacteria were isolated and purified from radish sprouts (Appendix A). Phylogenetic analysis showed that the bacteria isolated from radish sprouts were mainly composed of proteobacteria, actinobacteria, and bacteroidetes, of which proteobacteria was the largest phylum (Figure 1).

### 3.2. Isolation and Identification of Gut Bacteria of P. xylostella 

Seven different strains of bacteria were identified from the gut bacteria of the 4th instar larvae of *P. xylostella* feeding on radish sprouts. Phylogenetic analysis showed that the bacteria isolated from the gut of the 4th instar larvae of *P. xylostella* were composed of proteobacteria and actinobacteria, of which proteobacteria was the largest (Figure 2A, Appendix A). Six different strains of bacteria were identified in the pupal gut, which was composed of proteobacteria, firmicutes, and actinobacteria (Figure 2B, Appendix A). Three different strains of bacteria were identified from the adult gut, composed of proteobacteria and actinobacteria (Figure 2C, Appendix A). A total of 12 strains of different bacteria were identified from the ovary, composed of proteobacteria and firmicutes, of which proteobacteria was the largest (Figure 2D, Appendix A). A total of 7 strains of different bacteria were identified from the eggs, composed of proteobacteria and firmicutes, of which proteobacteria was the largest (Figure 2E, Appendix A). The results showed that the largest phylum of gut symbiotic bacteria of *P. xylostella* is proteobacteria.

### 3.3. Correlation Analysis between Gut Bacteria of P. xylostella and Food 

The common bacteria found through culturing the radish sprouts, the gut of different stages of larvae, ovary, and eggs of *P. xylostella* were used for phylogenetic analysis. The results showed that the bacteria of the same genus from different sources were clustered in the same branch and closely related, which may be the same species of bacteria (Figure 3). After comparing the bacteria belonging to the same genus in the larval gut, pupal gut, adult gut, ovary, and eggs of *P. xylostella* that feed on radish sprouts, it was found that these bacteria have a high degree of homology (Table 1). Previous studies suggested that the 16S rDNA sequence identity of the bacteria was more than 97%, which could be considered the same species [25]. Sequence analysis showed that the bacteria of the same genus isolated from radish sprouts and the gut, ovary, and eggs of *P. xylostella* could be considered the same bacteria (Table 1).

Proteobacteria and actinobacteria were the main bacteria in the gut of *P. xylostella* feeding on radish sprouts (Appendix A). In proteobacteria, the same *Enterobacter* sp., *Pantoea* sp., and *Cedecea* sp. were found in the 4th instar larval gut of *P. xylostella* and radish sprouts (Appendix A). In addition, the gut bacteria of *P. xylostella* feeding on radish sprouts and artificial diet were significantly different on the LB medium (Appendix A). These results indicated that the gut bacteria of *P. xylostella* are potentially related to the food it eats. 

The bacteria in the gut, ovaries, and eggs of *P. xylostella* were mainly composed of bacteria from the phyla proteobacteria and firmicutes (Appendix A). *P. xylostella* had the same *Enterobacter* sp. in its gut, ovaries, and eggs. The ovaries and the 4th instar larval gut of *P. xylostella* had the same *Enterobacter* sp., *Pantoea* sp., and *Cedecea* sp., the ovaries and eggs had the same *Enterobacter* sp., *Carnobacterium* sp, and *Lysinibacillus* sp. (Appendix A). These indicated that the gut bacteria of *P. xylostella* may be transferred to the ovary, and the ovary to the egg, to realize the vertical transmission of gut bacteria of *P. xylostella*.

### 3.4. Analysis of Vertical Transmission of Gut Bacteria from P. xylostella

*Enterobacter* sp. RE1-KN has kanamycin resistance and can be used as an indicator for screening and identification. The experiment found that there was no *Enterobacter* sp. RE1-KN in the gut, ovary, and egg surface of *P. xylostella* feeding with a normal diet, while *Enterobacter* sp. RE1-KN was detected in the 4th instar larval gut, pupal gut, adult gut, ovary, and egg surface of *P. xylostella* feeding with a diet containing *Enterobacter* sp. RE1-KN (Figure 4B,C). The results indicated that the gut bacteria of *P. xylostella* can be transmitted to the ovaries and eggs.

It was found that the presence of *Enterobacter* sp. RE1-KN was not detected in the gut of the 4th instar larvae developed from eggs that were soaked with sterile water, while the presence of *Enterobacter* sp. RE1-KN was detected in the gut of the 4th instar larvae which developed from eggs soaked with *Enterobacter* sp. RE1-KN solution (Figure 4D,E). The results showed that the bacteria on the egg surface can spread to the gut of *P. xylostella*.

### 3.5. Analysis of Horizontal Transmission of Gut Bacteria from P. xylostella

In mixed feeding of *P. xylostella* with and without gut bacteria, the survival rates of *P. xylostella* in four replicates were 70%, 70%, 60%, and 90%, respectively. Importantly, *Enterobacter* sp. RE1-KN was detected in the gut of all surviving *P. xylostella* (Appendix A and Figure 5B). The results showed that gut bacteria of *P. xylostella* can be horizontally transmitted within populations through social activities.

## 4. Discussion

Whether feeding on rice or maize, proteobacteria was found to be the largest phylum in the gut of *Cnaphalocrocis medinalis* [26]. Similarly, *Bactrocera minax* collected from a vegetable field had proteobacteria as the largest phylum in its gut [27]. Furthermore, the largest phylum of gut bacteria in *P. xylostella* feeding on radish sprouts is also proteobacteria in this study. These show that proteobacteria are widely present in the gut of insects that feed on natural food. Moreover, this study showed that the gut bacteria of *P. xylostella* are potentially related to the food it eats. These indicate a potential correlation between insect gut bacteria and the food they consume, the phenomenon shared with *S. frugiperda*, *Bactrocera dorsalis,* and *Nezara viridula* [8,28,29,30]. However, studies have shown that both environmental factors and food can affect insect gut microbial diversity [20]. For example, the gut bacteria of *P. xylostella* can be affected by different geographical regions [19], and there are significant differences in the bacterial community structure of the gut of *Musca domestica* under field and laboratory conditions [31]. In addition, the gut microflora structure of insects may change during different life stages. For example, the diversity of the gut bacterial community of the larvae of *Gastrolina depressa* is generally higher than that of adults, and the diversity of the gut bacterial community of 1st and 2nd instar larvae is the highest [32]. Therefore, the diversity of the gut bacterial community of insects is affected by food, environment, and their life history, these factors may work together in the construction of gut bacterial community diversity, affecting the growth and development of insects.

Our previous study also found that different host plants can affect the diversity of gut bacteria of *P. xylostella* [33]. This study found that many bacteria in food can be transferred to the gut of *P. xylostella* by the traditional culture method, but this method is based on the similarity of the 16S rDNA of isolated bacteria. Although the similarity of most bacteria is more than 99%, based on the current study, the bacterial taxonomic units have entered the level of strains, and these bacteria with highly similar 16S rDNA belong to the same genus; however, they may belong to different strains [34]. Therefore, it is necessary to study the correlation between gut bacteria of *P. xylostella* and food more accurately and systematically by bacterial markers and other methods in the future.

In addition, we found that the gut bacteria of *P. xylostella* can be transmitted to ovaries and eggs, and the bacteria carried by eggs can further spread to the next generation. The gut bacteria of *P. xylostella* have a route of vertical transmission through the eggs. This phenomenon is similar to *Tribolium castaneum*, where Knorr et al. fed *T. castaneum* with fluorescent-labeled *Escherichia coli* and *Pseudomonas entomophila*, and traced the labeled bacteria in the female reproductive system and eggs of *T. castaneum* [35]. The bacterial species *Serratia symbiotica* was originally characterized as noncultured strains that live as mutualistic symbionts of *Aphidoidea* and are vertically transmitted through transovarial endocytosis within the mother’s body [11]. *Snodgrassella alvi* and *Gilliamella apicola* in *Bombus terrestris* populations can also be vertically transmitted from the mother to the offspring [10], suggesting that vertical transmission of gut bacteria through eggs is likely a common phenomenon in insects. However, although this study confirmed that eggs carrying *Enterobacter* sp. RE1-KN can transmit to the offspring, further investigations are needed to examine its stability inside the eggs and in the gut of *P. xylostella* after multiple generations. Previous studies have shown that some social insects, such as *Cryptocercus* sp., *R. speratus,* and *A. mellifera*, can horizontally transmit gut bacteria through population activities, such as trophallaxis or coprophagy [12,13,14]. This study found that the larvae of *P. xylostella* can also carry out the horizontal transmission of gut bacteria within the population. Currently, trophallaxis of *P. xylostella* has not been observed. On the contrary, in the case of food shortage, there was a very serious phenomenon of cannibalism. In this experiment, an adequate amount of food was provided, so cannibalism was not observed. Therefore, the horizontal transmission of gut bacteria in *P. xylostella* may occur through feces excreted by the larvae, which are transmitted to other *P. xylostella* living in the same space through food transfer. Another possibility is that during the feeding process of *P. xylostella*, its oral regurgitation fluid may also contain some gut bacteria, which can also be left on the surface of food and help establish gut microbiota in subsequent feeding by other *P. xylostella*. However, these specific forms and mechanisms need further determination. This result also suggests that group-living insects, not just social insects, may experience the horizontal transmission of gut bacteria due to the effects of regurgitation fluid and feces when feeding in the same space. In addition, previous studies have found that the horizontal transmission of gut bacteria in some insect species can occur not only within populations but also between populations, such as the gut flora of *Xylocopa micans* having high homology with those of *A. mellifera* and *Bombus ruderarius* [36]. Whether the gut bacteria of *P. xylostella* can also carry out horizontal transmission among populations with its homologous species or species with the same host needs further study.

In addition, the engineered bacterium *Enterobacter* sp. RE1-KN, which was constructed to study vertical and horizontal transmission in this study contains a plasmid with the green fluorescent protein (GFP) gene. Our original purpose was to better trace and display the whole process of gut microbial transmission in vivo through fluorescent labeling. However, it is possible that the host bacterium, *Enterobacter* sp. RE1, lacks the transcription factors necessary for GFP expression from the PET28a-EGFP plasmid but possesses transcription factors containing the kanamycin resistance gene. Therefore, in this study, only kanamycin resistance markers were used to track bacterial migration. Future studies can further construct engineered bacteria that can stably express fluorescent protein for more convenient and visual exploration. The fluorescent labeling and high-throughput sequencing methods can be further combined to study whether the labeled bacteria still exist in the egg and at each growth stage after multiple generations of transmission, to evaluate the stability of this transmission mode, and further study the molecular mechanism of vertical and horizontal transmission. Another point of concern is that the purpose of using an anaerobic agar medium in this paper was to increase the number of bacteria screened, but interestingly, all the bacteria screened on this medium were facultative anaerobes rather than strict anaerobes. This may be due to the relatively straight gut structure of *P. xylostella*, which is not easy to form a closed anaerobic space in a certain area. It may also be because this anaerobic device can not completely exclude oxygen; therefore, strict anaerobes have not yet been isolated. In the future, more advanced anaerobic culture devices can be used to explore the composition and function of anaerobic bacteria in the gut of *P. xylostella*. Of course, the extensive presence of facultative anaerobes in the gut of *P. xylostella* in this study also shows the adaptability of such bacteria to the semi-closed structure of the gut from another perspective. Another disadvantage of this study is that based on the current data, it is not clear whether the main source of gut bacteria of *P. xylostella* is food or its vertical and horizontal transmission. We only know that *P. xylostella* can establish gut flora through these three ways, but which is the main way? What is the proportion of the three modes of transmission? These problems cannot be quantified at present, and these problems are related to the establishment and use of biological control methods based on gut bacteria in the future. Therefore, more quantitative experiments need to be designed to determine the main source of gut bacteria in *P. xylostella*. In addition, this study did not explore which social activities were involved in the horizontal transmission of gut bacteria by *P. xylostella*. In the future, it is necessary to explore the role of regurgitation fluid and feces from *P. xylostella* in the horizontal transmission of gut bacteria.

Finally, in microbial research, insects are subjected to aseptic treatment before being dissected, but the aseptic-treated insect body is not usually sampled for the detection of bacteria, as we generally believe that bacteria on the surface of insects soaked in alcohol and pure water will be killed. However, in order to ensure the high reliability of the research, subsequent microbiological studies require the sterile validation of sterilized insect bodies. Additionally, in this study, all PCR experiments were conducted using a culture medium containing kanamycin, the recombinant bacterium *Enterobacter* sp. RE1-KN with kanamycin resistance was first isolated from *P. xylostella* by the selective medium, and then PCR amplification was performed using the specific primer T7 of the recombinant plasmid. The length of the primer also met theoretical expectations, and the entire experiment strictly followed aseptic procedures, so the contamination can be ruled out in theory. However, from a rigorous experimental perspective, negative controls should be added to make the experiment more rigorous. In future studies, we will add both negative and positive controls to improve the reliability and rigor of the experiment.

## 5. Conclusions

The gut bacteria of *P. xylostella* is related to its food. *P. xylostella* can obtain these bacteria from its diet to establish its gut flora and transmit the bacteria to the next generation via the ovary and egg. In addition, the gut bacteria of *P. xylostella* can be vertically transmitted through eggs and horizontally transmitted within the population. This study laid a foundation for further research on the gut bacteria of *P. xylostella* in the future and provided a new idea for the control of *P. xylostella* from the perspective of the source and transmission modes of gut bacteria.

## Figures and Tables

**Figure 1 insects-14-00504-f001:**
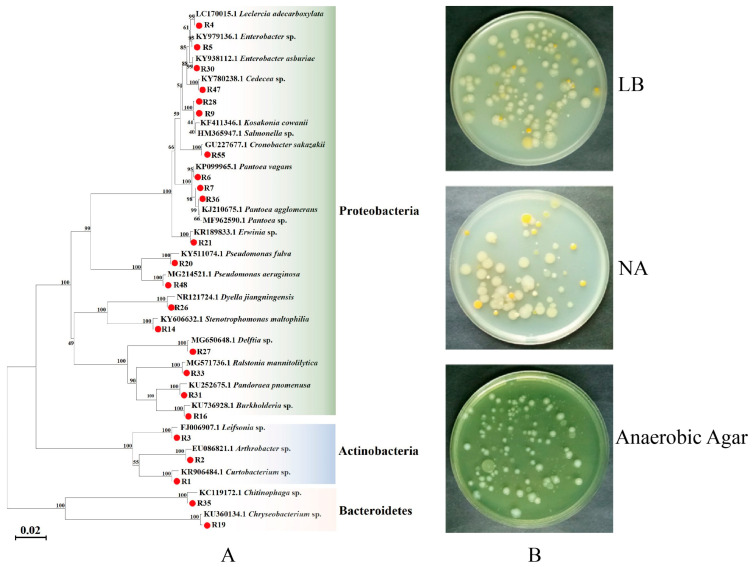
Phylogenetic analysis of 16S rDNA of bacteria from (**A**) neighbor-joining tree of bacterial isolates from radish sprouts and their closely related species based on sequencing of the 16S rDNA gene. The numbers corresponding to the red circle in the figure are the bacteria obtained in this study. The nodes’ bootstrap values were based on 1000 replicates. The scaled bar represents 0.02 estimated phylogenetic divergence. (**B**) Isolation and culture of radish sprout bacteria on Luria-Bertani medium (LB), nutrient agar (NA), and anaerobic agar solid mediums.

**Figure 2 insects-14-00504-f002:**
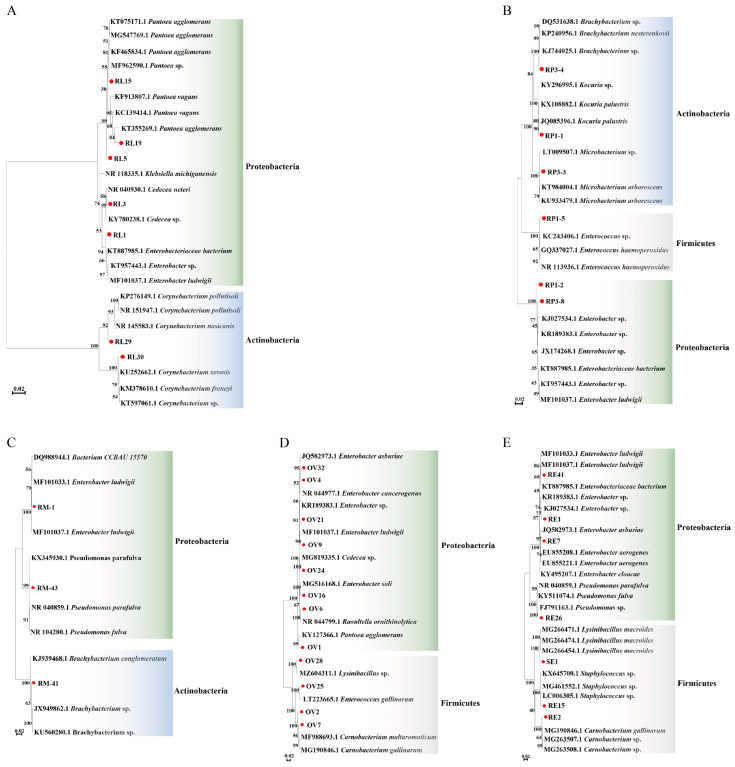
Neighbor-joining tree of bacterial isolates from *P. xylostella* and their closely related species based on sequencing of the 16S rDNA gene. The nodes’ bootstrap values were based on 1000 replicates. The scaled bar represents 0.02 estimated phylogenetic divergence. (**A**–**E**), respectively, represent phylogenetic analysis of 16S rDNA of bacteria of the 4th instar larval gut, pupal gut, adult gut, ovary, and eggs in *P. xylostella*. The numbers corresponding to the red circle in the figure are the bacteria obtained in this study: R represents bacteria in radish sprouts, RL represents gut bacteria in the 4th instar larvae of *P. xylostella*, RP represents gut bacteria in the pupae of *P. xylostella*, RM represents gut bacteria in adult *P. xylostella*, OV represents ovarian bacteria in *P. xylostella*, and SE and RE represent bacteria in eggs of *P. xylostella*.

**Figure 3 insects-14-00504-f003:**
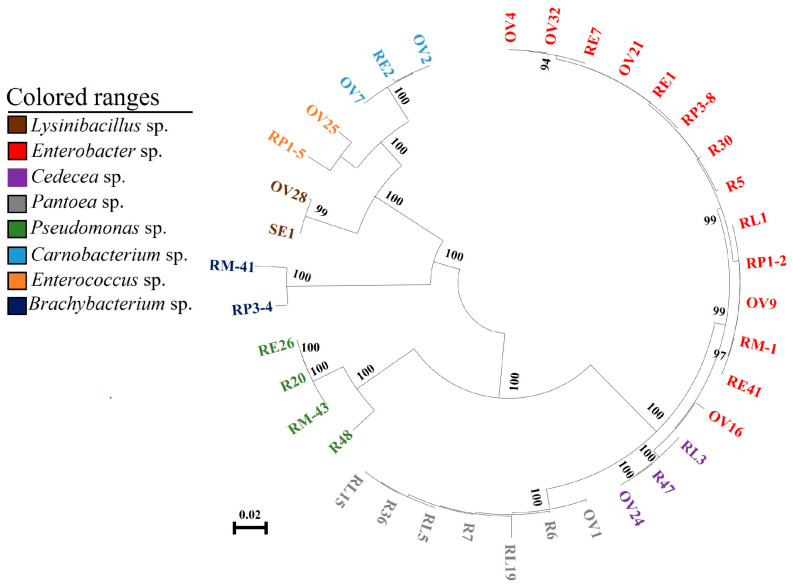
Neighbor-joining tree of bacterial isolates from *P. xylostella*. The nodes’ bootstrap values were based on 1000 replicates. The scaled bar represents 0.02 estimated phylogenetic divergence. R represents bacteria in radish sprouts, RL represents gut bacteria in the 4th instar larvae of *P. xylostella*, RP represents gut bacteria in the pupae of *P. xylostella*, RM represents gut bacteria in adult *P. xylostella*, OV represents ovarian bacteria in *P. xylostella*, and SE and RE represent bacteria in eggs of *P. xylostella*. As shown in the colored range in the figure, different colors indicate different genera.

**Figure 4 insects-14-00504-f004:**
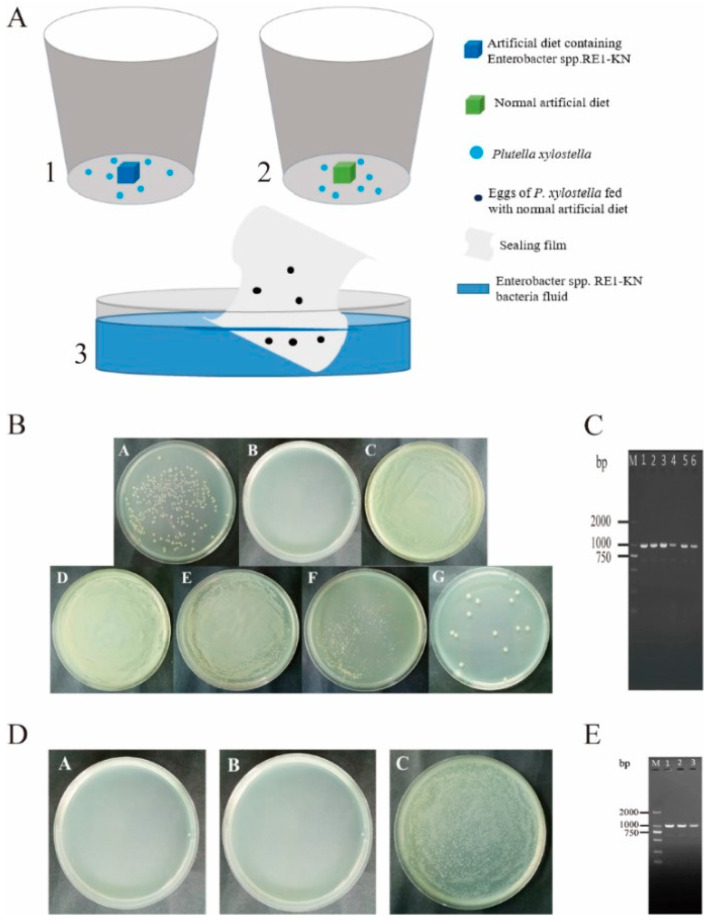
Validation of vertical transmission of gut bacteria in *P. xylostella*. (**A**) Experimental method of vertical transmission of bacteria in the gut of *P. xylostella*. (A-1) *P. xylostella* fed with a diet containing *Enterobacter* sp. RE1-KN, (A-2) *P. xylostella* fed with a normal diet, (A-3) the sealing film with *P. xylostella* eggs was soaked in *Enterobacter* sp. RE1-KN solution; (**B**) isolation and culture of bacteria from different stages of *P. xylostella*. (B-A) *Enterobacter* sp. RE1-KN, (B-B) Control, (B-C) gut of 4th instar larvae, (B-D) pupal gut, (B-E) adult gut, (B-F) adult ovary, (B-G) egg surface. (**C**) *Enterobacter* sp. RE1-KN was detected in 4th instar larval gut, pupal gut, adult gut, ovary, and egg surface of *P. xylostella* feeding with diet containing *Enterobacter* sp. RE1-KN. (C-M) DL2000 DNA Marker, (C-1) *Enterobacter* sp. RE1-KN, (C-2) Gut of 4th instar larvae, (C-3) pupal gut, (C-4) adult gut, (C-5) adult ovary, (C-6) egg surface. (**D**) Isolation and culture of gut bacteria from 4th instar larvae of *P. xylostella*. (D-A) Sterile water for the last cleaning of eggs: eggs of *P. xylostella* were sterilized with 1.5% sodium hypochlorite and the sterile water used to clean the eggs for the last time was tested to be free of bacteria with an LB medium, (D-B) gut of the 4th instar larvae were developed from eggs soaked in sterile water, (D-C) gut of the 4th instar larvae which were developed from eggs soaked in *Enterobacter* sp. RE1-KN solution. (**E**) The presence of *Enterobacter* sp. RE1-KN was detected in the gut of the 4th instar larvae which developed from eggs soaked with *Enterobacter* sp. RE1-KN solution. The PCR amplification (E-1,E-2,E-3) of *Enterobacter* sp. RE1-KN in the gut of the 4th instar larvae which developed from eggs soaked in *Enterobacter* sp. RE1-KN solution, (E-M) DL2000 DNA Marker.

**Figure 5 insects-14-00504-f005:**
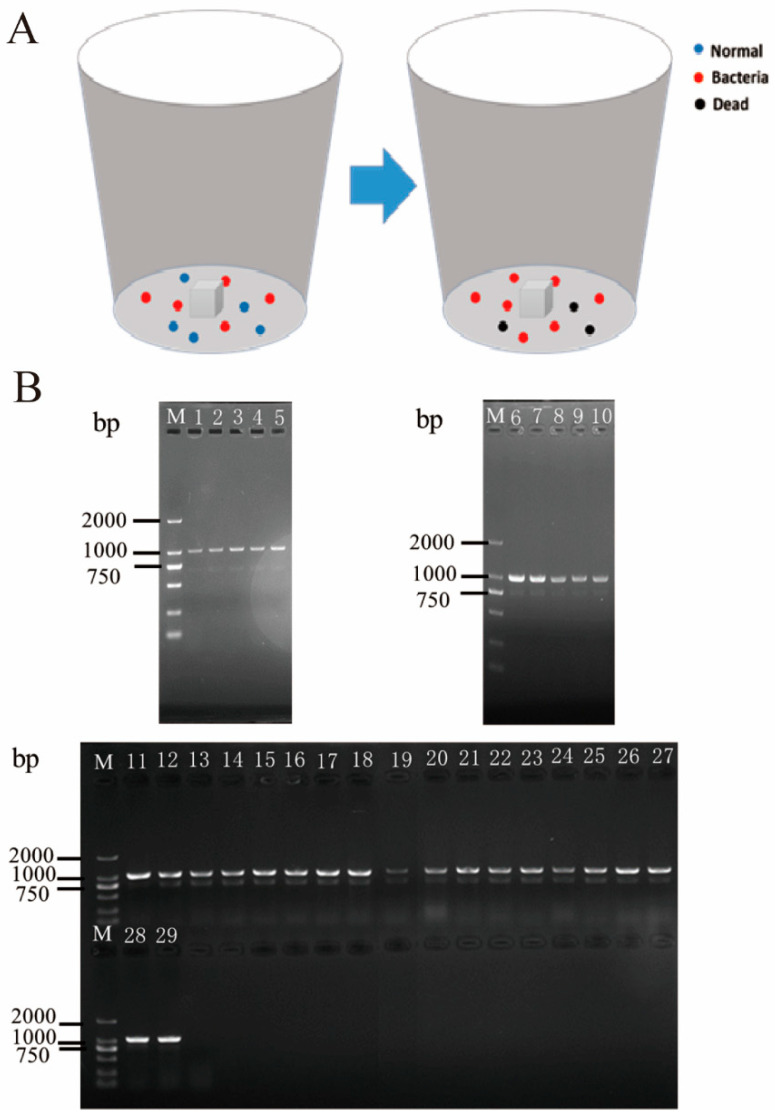
Validation of horizontal transmission of gut bacteria in *P. xylostella*. (**A**) The 1st instar larvae of *P. xylostella* were fed with a diet containing *Enterobacter* sp. RE1-KN as the treatment group, while they were fed a normal diet, the same as the control group, and they were raised to the 3rd instar larvae, respectively. Then, the 3rd instar larvae of the control group and treatment group were mixed and reared together as shown in the insect-rearing container on the left, and then reared together until the 4th instar as shown in the container on the right. They were fed on the normal diet when mixed. The experiment was repeated four times. Blue dots represent normal *P. xylostella*, red dots represent *P. xylostella* containing *Enterobacter* sp. RE1-KN, and black dots represent dead *P. xylostella*. (**B**) The PCR amplification of *Enterobacter* sp. RE1-KN in the gut of 4th instar larvae which were developed from the mixed rearing of the 3rd instar larvae. M: DL2000 DNA Marker, 1–29: detection of *Enterobacter* sp. RE1-KN in each surviving *P. xylostella* gut.

**Table 1 insects-14-00504-t001:** Comparison of bacteria of the same genus from different sources.

Strain	Sequence Alignment			Sources	
	Sequence A	Sequence B	Identity	B	A
*Enterobacter* sp.	R5	R30	99%	Radish sprouts	Radish sprouts
		RL1	99%	Larval gut	
		RP1-2, RP3-8	99%	Pupal gut	
		RM-1	99%	Adult gut	
		OV4, OV9, OV21, OV32	99%	Ovary	
		OV16	98%	Ovary	
		RE1, RE7, RE41	99%	Egg	
*Pantoea* sp.	R6	R7, R36	99%	Radish sprouts	Radish sprouts
		RL5, RL15	99%	Larval gut	
		RL19	98%	Larval gut	
		OV1	99%	Ovary	
*Cedecea* sp.	R47	RL3	100%	Larval gut	Radish sprouts
		OV24	100%	Ovary	
*Pseudomonas* sp.	R20	R48	95%	Radish sprouts	Radish sprouts
		RM-43	99%	Adult gut	
		RE26	100%	Egg	
*Brachybacterium* sp.	RP3-4	RM-41	97%	Adult gut	Pupal gut
*Enterococcus* sp.	RP1-5	OV25	97%	Ovary	Pupal gut
*Carnobacterium* sp.	OV2	OV7	99%	Ovary	Ovary
*Lysinibacillus* sp.	OV28	SE1	99%	Egg	Ovary

Notes: in the table, sequence B is compared with sequence A. Sources A and B are the tissues from which the sequences A and B were isolated; R represents bacteria in radish sprouts, RL represents gut bacteria in the 4th instar larvae of *P. xylostella*, RP represents gut bacteria in the pupae of *P. xylostella*, RM represents gut bacteria in adult *P. xylostella*, OV represents ovarian bacteria in *P. xylostella*, and SE and RE represent bacteria in eggs of *P. xylostella.*

## Data Availability

The data presented in this study are available on request from the corresponding author. The data are not publicly available due to restrictions, e.g., privacy or ethics.

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
