# Peer review of "Potential Source and Transmission Pathway of Gut Bacteria in the Diamondback Moth, Plutella xylostella"

_insects, 2023, doi:10.3390/insects14060504_

Round 1

Reviewer 1 Report

 Potential source and transmission pathway of gut bacteria in the diamondback moth, Plutella xy-lostella

Shuncai Han1,2,3†, Qianqian Ai1,2,3†, Xiaofeng Xia1,2,3

All the manuscript needs an English revision. Reading is not fluid and difficult to understand due to grammatical errors. There are also misspellings with dash in the middle of words and punctuation errors in sentences. And proofreading would be easier if the lines were numbered.

Correct the writing of all scientific species name to italic, ex: Plutella xylostella, P. xylostella. Also write the genera of bacteria as it should be in a scientific publication including sp. or spp. and formatting to italic.

Abstract:

Authors should mention the main genera of cultivable bacteria found.

Detail the horizontal transmission. That it occurs through the insect’s social behavior.

Introduction:

First paragraph: specify that Ponkan and Shatangju are citrus cultivar.

 Second paragraph: include sp. after the bacteria gender: Snodgrassella and Gilliamella

 Material and Methods:

2.1. Feeding P. xylostella

What is Vc: 1 g Vc ??

Why the relative humidity so different when comparing the feeding on artificial diet and radish sprout?

What was the stage of development of P. xylostella used in different feeding diets?

2.2.1. Isolation and culture of bacteria from radish sprout

How many days after seeding was the radish sprout leaves? What was the size of the leaves?

The medium protocol is missing information.

The dilution of radish sprout was ten to the power of minus one (10-1, 10-2 and 10-3)?

What was the dissolved solution?

2.2.2. Isolation and culture of gut bacteria from P. xylostella

It was used 30 insects of each developmental stage or 10 of each three stages (larvae, pupae and adults)?

The authors did not detail the dissection of 4th instar larvae and pupae. Only adults.

Correct the dilutions spelling.

No detail of statical analyses is presented.

Results:

Figure legends needs more details, such as abbreviations, number of biological and experimental repetition, and statistical analysis.

The authors did not submit the supplementary data.

Some figures do not bring all the details needs to reach to the conclusion.

Figure 1 does not give the readers any special information. Maybe if it was associated with the data presented in the supplementary table.

Figure 3 should be clearer about which figure letter is related to which organ used as sample.

Table 1 – it is difficult to understand the rows of each bacteria genus.

There is no explanation of the abbreviations.

Discussion:

The discussion is superficial. Needs to improve. The authors need to present a more in-depth discussion on the manuscript results, in particular related with the bacteria found.

Reviewer 2 Report

In my opinion, the manuscript should be strongly improved. In this state, it should not be accepted. here are my major comments:

1. English language must be improved. Numerous grammar mistakes were found.

2. The first paragraph of Introduction must be changed because it contains too many details.

3. How does the diet resemble the natural one?

4. 2.2.2. 30 larvae, 30 pupae and 30 adults or the number of 30 concern the sum of specimens?

5. Why the 4th larval stage was chosen? it is not explained.

6. Authors should describe in the introduction all larval stages of analyzed species.

7. How long does embryogenesis persist? Were all the eggs used in studies at the same stage of development? How did you check it?

8. Results - I can't find any information about the type of ovary.

9. Results - 3.4. and 3.5. methods should be written in the Methodology section, not in the Results.

10. When Authors delete methodology from chapters 3.4 and 3.5, only 2-3 sentences would form these sections.

11. Conclusions - I can't find any conclusions. This chapter resembles the Abstract.
